# Contrast Clearance Analysis (CCA) to Assess Viable Tumour following Stereotactic Radiosurgery (SRS) to Brain Metastasis in Non-Small Cell Lung Cancer (NSCLC)

**DOI:** 10.3390/cancers16061218

**Published:** 2024-03-20

**Authors:** Shybi Mohamedkhan, Sumeet Hindocha, James de Boisanger, Thomas Millard, Liam Welsh, Philip Rich, Andrew D. MacKinnon, Nicholas Williams, Bhupinder Sharma, Nicola Rosenfelder, Anna Minchom

**Affiliations:** 1Royal Marsden Hospital, Downs Rd., Sutton SM2 5PT, UK; 2Royal Marsden Hospital, Fulham Rd., London SW3 6JJ, UK; 3Institute of Cancer Research, London SW7 3RP, UK; 4Imperial College London, London W12 0NN, UK

**Keywords:** contrast clearance analysis, non-small cell lung cancer, brain metastases, radionecrosis, stereotactic radiosurgery

## Abstract

**Simple Summary:**

Brain metastases are common in lung cancer and increasingly treated using targeted stereotactic radiosurgery (SRS). Post-SRS changes (including radionecrosis) may be difficult to distinguish from progressive brain metastasis on MRI. This can have important implications for guiding the most appropriate further management. Contrast clearance analysis (CCA) presents an alternative imaging technique to aid differentiation of progressive tumour from post-treatment changes. In this study, we evaluate the role of CCA, assessing its utility in a real-world setting of patients with NSCLC treated for brain metastases. In particular, we assess the impact of CCA interpretations on treatment decisions and the effects of using this imaging technique with suggestions for best practices. Our experience shows CCA to be feasible and useful in patients with NSCLC in cases of diagnostic uncertainty in MRI. Recommendations include the appreciation of a thin rim of rapid contrast clearance, which can be seen in responding tumours, and the importance of the timing of CCA prior to surgery for suspected brain metastasis progression.

**Abstract:**

Background and Objective: Brain metastases are common in lung cancer and increasingly treated using targeted radiotherapy techniques such as stereotactic radiosurgery (SRS). Using MRI, post-SRS changes may be difficult to distinguish from progressive brain metastasis. Contrast clearance analysis (CCA) uses T1-weighted MRI images to assess the clearance of gadolinium and can be thus used to assess vascularity and active tumours. Design and Methods: We retrospectively assessed CCAs in 62 patients with non-small cell lung cancer (NSCLC) undergoing 104 CCA scans in a single centre. Results: The initial CCA suggested the aetiology of equivocal changes on standard MRI in 80.6% of patients. In all patients whose initial CCA showed post-SRS changes and who underwent serial CCAs, the initial diagnosis was upheld with the serial imaging. In only two cases of a presumed progressive tumour on the initial CCA, subsequent treatment for radionecrosis was instigated; a retrospective review and re-evaluation of the CCAs show that progression was reported where a thin rim of rapid contrast clearance was seen, and this finding has been subsequently recognised as a feature of post-treatment change on CCAs. The lack of concordance with CCA findings in those who underwent surgical resection was also found to be due to the over-reporting of the thin blue rim as disease in the early cases of CCA use and, in three cases, potentially related to timelines longer than 7 days prior to surgery, both factors being unknown during the early implementation phase of CCA at our centre but subsequently learned. Conclusions: Our single-centre experience shows CCA to be feasible and useful in patients with NSCLC in cases of diagnostic uncertainty in MRI. It has helped guide treatment in the majority of patients, with subsequent outcomes following the implementation of the treatment based on the results, suggesting correct classification. Recommendations from our experience of the implementation include the careful consideration of the thin rim of the rapid contrast clearance and the timing of the CCA prior to surgery for suspected brain metastasis progression.

## 1. Introduction

Brain metastases are detected in 10–20% of patients with non-small cell lung cancer (NSCLC) at the time of diagnosis [1,2], with a higher incidence in patients with certain molecular subtypes [3]. Approximately 50–60% of patients with the *EGFR* mutant or *ALK* fusion-positive NSCLC develop brain metastases during the course of their disease [4,5,6] and this may increase as extracranial disease control improves with better systemic therapy [3,7,8]. Systemic therapy agents have variable intracranial penetrance [9,10,11] or are effective for a limited time and local treatment with surgical resection and/or radiation is often required [12,13]. Stereotactic radiosurgery (SRS) has become the standard of care for patients with a limited volume of intracranial disease not suitable for surgical resection, delivering high-dose radiation in a single or few fractions, with rapid dose fall-off outside the target, and improving survival with lower toxicity compared with whole-brain radiotherapy (WBRT) [14,15,16].

Radionecrosis is the foremost late side effect of SRS and is estimated to occur in 5–25% of cases [17,18,19]. Although often asymptomatic, symptoms occur in approximately half of those with radiological changes, including headache, fatigue, nausea, imbalance, weakness and seizures [20]. Radionecrosis risk increases with the prescribed dose, fraction size, treated volume, previous irradiation, previous surgery to the treated site and the use of concurrent systemic therapy [13,17,18,19]. Treatment options for radionecrosis include surveillance, corticosteroids, bevacizumab and surgical resection [13,21,22,23]; Laser Interstitial Thermal Therapy (LITT) [24], pentoxiphylline and vitamin E may be of benefit although require further assessment [25]. However, treatment for progressive disease includes further surgery, re-irradiation or a change in systemic treatment—the latter two being inappropriate for radionecrosis.

Contrast-enhanced magnetic resonance imaging (MRI) remains the imaging modality of choice for brain metastases. However, the changes in radionecrosis (contrast enhancement, perilesional oedema and temporal changes) mimic those of progressive disease and following SRS, it can be difficult to differentiate a viable tumour from radionecrosis [13]. Although tissue diagnosis remains the gold standard, this may not be feasible or appropriate due to the associated risks [13]. MR perfusion, MR spectroscopy and PET-CT have been proposed to aid diagnosis, but no modality has proved superior to contrast-enhanced MRI [13].

Contrast clearance analysis (CCA) presents an alternative imaging technique to aid in the differentiation of progressive tumours from post-treatment changes [26]. T1-weighted MRI sequences are acquired at 5 min and again at >60 min after gadolinium administration. A colour map is obtained by subtracting the early post-contrast images from the late post-contrast scan using dedicated Brainlab Elements software (Brainlab, Munich, Germany). The map demonstrates the handling of the contrast between the two scans [27]; blue regions represent the rapid clearance of the contrast (in which the delayed signal is less than the early signal), suggestive of an active tumour, whereas red regions represent contrast accumulation (in which the delayed signal is greater than the early signal) due to damaged vasculature, indicative of treatment-related damage, or radionecrosis. CCA is reported to have 100% sensitivity and a 92% positive predictive value for active tumours [26] and compares favourably with MR perfusion [28].

In this study, we evaluate the role of the CCA, assessing its utility in a real-world setting of patients with NSCLC treated for brain metastases. In particular, we assess the impact of CCA interpretations on treatment decisions and the effects of using this imaging technique with suggestions for best practices.

## 2. Methods

We performed a retrospective single-centre study of patients with NSCLC treated at RMH for brain metastases who received a CCA MRI between 1 January 2018 and 1 March 2022. Approval to use anonymised data was granted by the Institutional Review Board (Royal Marsden Committee for Clinical Research, ref:SE1108) and did not require patient consent. CCAs are performed according to clinical need, where changes in contrast-enhanced MRI are equivocal.

A list of patients with NSCLC who underwent CCA post-SRS was extracted from a prospectively collected database of all patients undergoing CCA. Patients were excluded if they were lost to follow-up. The electronic patient record system was manually reviewed to collect data for each patient: date of birth, sex, date of NSCLC diagnosis, date of brain metastases diagnosis, NSCLC histology, molecular drivers (if any), date of death (if applicable), systemic anticancer treatment (SACT) received, treatment for brain metastases, histological report following surgical resection of brain metastases (if applicable), number, date and report of CCA scans, concordance of CCA report and histology from surgical resection (if applicable) and treatment decisions following CCA. The CCA was mapped using Brainlab Elements post processing software (Brainlab, Munich, Germany) and reported by neuro-radiologists as per our standard practice. A statistical analysis including an analysis of survival was performed using Microsoft Excel v16.81. Survival was analysed from the date of the histological diagnosis of NSCLC and the time of treatment of brain metastases and the median and range described.

## 3. Results

### 3.1. Demographics and Treatment

A total of 62 patients were included in this study; see Table 1 for baseline characteristics.

Patients received up to six lines of SACT during the study period, comprising either chemotherapy, targeted treatment with a tyrosine kinase inhibitor (TKI) or immunotherapy (Table 1).

Twenty-three patients (37%) had brain metastases at diagnosis. Of the 39 that did not, the median time from diagnosis to the detection of brain metastases was 13 months (range 1–120). Brain-directed therapy is detailed in Table 1. The median time from diagnosis to the first treatment for brain metastases was 10 months (range 0–132).

### 3.2. Contrast Clearance Analysis

Over the time period assessed, 104 CCA scans were performed with a median of one scan per patient (range of 1–7). A total of 21 patients had more than one CCA. The first CCA was performed at a median of 12.7 months (range 2.0–71.5) following SRS/WBRT treatment. The median overall survival of the patient cohort from the time of NSCLC diagnosis was 71.0 months and 29.0 from the date of the first SRS/WBRT.

The first performed CCA demonstrated post-radiotherapy changes in 21 patients (Figure 1). Of these, three had received WBRT with 20 Gy in five fractions. The remainder had SRS at dose-fractionation schedules ranging between 17 and 24 Gy in a single fraction to 21–24 Gy delivered in three fractions. The median time from SRS/WBRT to the first CCA in this group was 9 months (range 1–54), and the median overall survival for this group was 72.0 months from the date of NSCLC diagnosis and 28.0 months from the date of the first SRS/WBRT. The recommended outcome was an ongoing CCA monitoring in all. Nine of these patients had between one and six further CCA scans, with post-SRS changes demonstrated in all follow-up scans except for one patient who had appearances in keeping with radionecrosis on their second CCA scan (performed 11 weeks after the first) but progressive tumour growth on their third (performed 13 weeks after the second). This patient subsequently underwent resection with histologically confirmed metastatic lung cancer. The remaining 12 patients are under monitoring and have not had further CCAs during the study period.

The first performed CCA suggested progressive tumour growth in previously treated SRS lesions in 20 patients. The median time from SRS to the first CCA was 9.0 months (range 0–34.0), and the median overall survival for this group was 58.0 months from the date of NSCLC diagnosis and 22.0 months from the date of SRS. The outcome was further monitoring in five patients (as further therapy was not deemed necessary immediately), further SRS in seven patients, WBRT in one patient, surgery with brain metastasis resection in four patients (one of which histologically confirmed metastatic lung cancer, three with radionecrosis), two had a change in systemic treatments to a different TKI and one patient was lost prior to follow up. The five patients having ongoing monitoring had between one and three further CCA scans with further brain active therapy initiated in three patients (one chemotherapy and two SRS). Two of the twenty patients originally thought to have disease progression on the CCA were subsequently treated for post-SRS changes; one patient had bevacizumab for post-SRS changes with improvement in imaging appearances and no subsequent disease progression and another was commenced on steroids for radionecrosis, suggesting the changes in the CCA were not due to progressive disease.

Eight patients had features of both post-SRS changes and tumour growth; in five patients, there were mixed features within the same lesion; in two patients there were mixed features across lesions; in one patient there were mixed features in the same lesion and across lesions. Five of these patients had one further CCA scan. Three of these CCAs showed clear tumours and the other two were consistent with post-radiotherapy changes. Of the eight patients with mixed appearances, one had further SRS, one had whole-brain radiotherapy and two had resection with histologically confirmed metastatic lung cancer. Two patients were commenced on steroids.

The first performed CCA demonstrated equivocal changes in 13 patients with ongoing CCA monitoring recommended in 12 and further SRS in 1. Eight of these patients had between one and four further CCA scans with four of these having tumours demonstrated on subsequent CCAs and two of these treated with SRS. The other four are under monitoring and have not had further CCAs during the study period.

In the five patients who underwent brain resection for what was thought to be progression on CCAs, the CCA was performed a median of 30 days prior to surgery. The CCA appearances in two patients, initially reported as tumours in 2017 (soon after the implementation of the CCA), were reclassified as treatment-related changes on a retrospective review concordant with histological findings; with increased experience, the neuro-radiologists recognised the changes in ‘a thin blue rim’ as being compatible with ongoing treatment response (Figure 2).

A case study demonstrating serial CCAs is shown in Figure 3.

## 4. Discussion

This case series represents, to our knowledge, the largest set of data of NSCLC patients undergoing CCAs. Others have reported CCAs in primary brain tumours and in a heterogeneous group of patients with brain metastases from a range of solid tumours [27].

The initial CCA suggested an answer as to the aetiology of the equivocal changes on their standard structural MRIs in 50 patients (80.6%). In the group with progressive tumours on the first CCA, the median survival from the time of radiation was 22 months compared with 28 months in those whose first CCA showed radiation-related change. Furthermore, in all patients whose initial CCA showed post-SRS changes and who underwent serial CCAs, the initial diagnosis was upheld with the serial imaging; two patients later developed progression which may be due to the absence of progression at early time points though the possibility that the CCA was less sensitive at assessing early disease progression cannot be excluded. Two patients of the twenty diagnosed with progressive tumours on the initial CCA were subsequently treated with bevacizumab or steroids, suggesting that the diagnosis may have been radionecrosis giving a positive predictive value of around 90%; the retrospective review and re-evaluation of early CCA reports show that progression was reported in some cases where a thin rim of rapid clearance was seen, a feature later recognised as being compatible with ongoing treatment-related change. This was not known at the start of CCA use and therefore some scans were reclassified following review at an international CCA multidisciplinary team meeting.

Overall, with the use of CCAs, 23 patients (37%) received treatment for progressive brain metastases (resection, radiotherapy, SRS, systemic therapy) and 5 (8%) received treatment for radionecrosis (steroids, bevacizumab). In those patients with equivocal CCAs reported and subsequent CCAs performed, all patients had a firm diagnosis made on the subsequent CCA and specific brain-directed treatment commenced as a consequence.

The assessment of the impact of CCAs on patient outcomes is not possible in this retrospective case series. However, it is demonstrated that this cohort of patients can receive brain-directed therapy whilst continuing to have extracranial control on or off systemic therapy with a long duration of median overall survival seen. This is of particular relevance to those with targeted therapy options due to their better survival, with the patients in this cohort with targetable aberrations being treated with TKIs for a median of 7 months (range < 1 to 52 months) since their first CCA scan.

Histological verification is the gold standard against which to test the validity of the CCA findings. Five patients underwent the resection of the presumed tumour, but only two had a viable tumour and three had radionecrosis on histology. The re-review of the preoperative imaging showed a “thin blue rim”, at the time thought to be suggestive of tumour progression but, with experience, recognised as more typical of radiation-related change. Additionally, we have learnt subsequently that CCA pre-surgery should be undertaken within 7 days of the operation (personal communication with Y Mardor and L Zach, Sheba hospital), and this was not known at the time of surgery for this small cohort and may have contributed to the non-concordance of CCA appearances and histology findings. Therefore, although the correlation of histology findings with the initial CCA report was low (40%), with the increased experience of CCAs, the correlation is higher, and at least 4/5 patients had features compatible with the CCA, -2/2 had tumours and at least 2/3 had treatment-related changes, even though the pre-op CCA was performed outside the recommended timeframe.

There are several limitations to this study. As this is a retrospective review, a comparison to other imaging modalities (e.g., CT or MRI alone) is lacking, as is the impact on patient outcomes. Also, the small number of patients undergoing the surgical resection of brain lesions limits the correlation of CCA findings with the histology. However, we demonstrate the practical impact of the incorporation of CCAs into clinical practice. Other than the processing software, no other equipment is required making it readily adoptable to any unit with an MRI scanner, and neuro-oncologist training is achievable in a short timeframe. The variation between tumours in their vascularity and contrast handling pre-treatment is recognised, and further work is being undertaken to understand these differences and the effect of these variations on treatment outcomes. A better understanding of the frequent scenario of mixed picture is required to know which changes are likely to herald radionecrosis and which are those of viable tumours, and in this latter group, to better differentiate ongoing tumour response (where rapid contrast clearance is seen but in a shrinking tumour) versus tumour progression. This work is being undertaken in a study assessing serial CCAs in patients pre- and post-SRS.

## 5. Conclusions

In this real-world data set, CCAs, interpreted in the context of routine structural MRI brain sequences, provided evidence to proceed to brain management in patients with NSCLC. Patients were maintained on systemic therapy for multiple lines of therapy. There is a learning curve for CCA interpretation and occasionally, post-SRS changes can vary and may resemble progressive disease. Pre-surgery timing and rim thickness should be incorporated in CCA guidelines in lung patients. Consensus input to develop guidelines for CCA use in lung cancer and prospective validation, including the impact on patient outcome, is warranted.

## Figures and Tables

**Figure 1 cancers-16-01218-f001:**
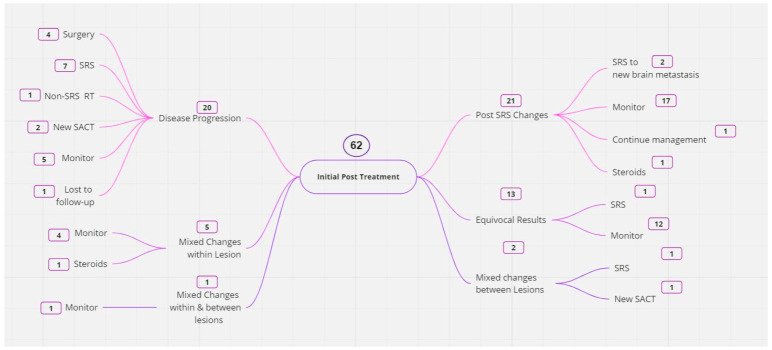
Outcomes following initial CCA scan (subsequent CCA outcomes not shown). SACT: systemic anti-cancer therapy, SRS: stereotactic radiosurgery.

**Figure 2 cancers-16-01218-f002:**
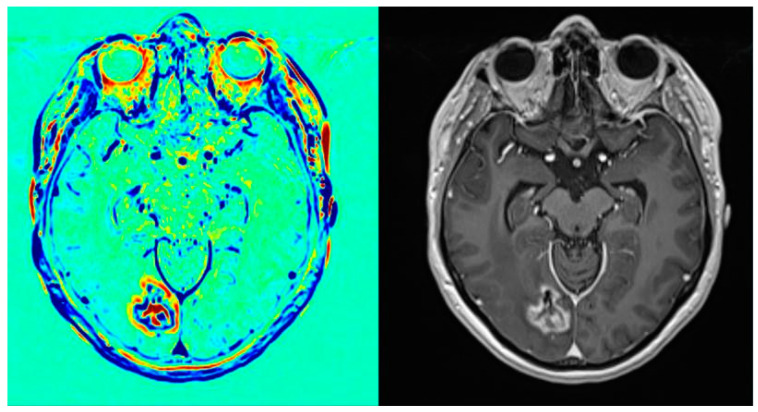
Case study of patient undergoing CCA demonstrating “thin blue rim” (arrowed) around tumour bed.

**Figure 3 cancers-16-01218-f003:**
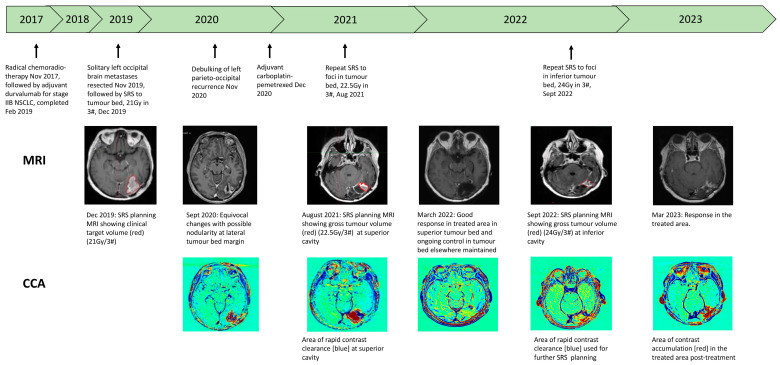
Case study of patient undergoing serial CCAs for suspected brain metastatic growth. CCA: contrast clearance analysis, Gy: grey, NSCLC: non-small cell lung cancer, SRS: stereotactic radiosurgery.

**Table 1 cancers-16-01218-t001:** Patient characteristics and treatments.

Factor	Patients Included in Study (*n* = 62)
Age, mean yrs	66, (range 33–90)
Female Gender, *n*	38 (61%)
Histology	
Adenocarcinoma:	54 (87%)
Squamous cell carcinoma:	4 (6.5%)
Not otherwise specified:	4 (6.5%)
Molecular profile (adenocarcinoma)	
No driver aberrations:	27 (43.5%)
*EGFR* mutation:	11 (17.7%)
*ALK* rearrangement:	11 (17.7%)
*KRAS* mutation:	10 (16.1%)
*ROS-1* rearrangement:	3 (4.8%)
Systemic treatments received over study period	
Chemotherapy only:	30 (48.4%) (range 1–3 lines)
Chemotherapy + PD-1/PD-L1 inhibitor:	20 (32.3%) (1–6 lines)
PD-1/PD-L1 inhibitor only:	1 (1.6%)
No SACT:	1 (1.6%)
ALK targeting:	11 (17.8%) (range 1–4 lines)
EGFR targeting:	9 (14.5%) (range 1–2 lines)
KRAS targeting:	1 (1.6%) (1 line)
ROS-1 targeting:	3 (4.8%) (range 1–2 lines)
Brain-directed therapy received over study period	
SRS alone:	35 (56.5%) (range 1–4 treatments)
SRS + WBRT:	3 (4.8%) (1–2 SRS treatment(s))
Surgical resection + SRS:	19 (30.6%) (1–6 SRS treatments)
Surgical resection + SRS + PBRT:	2 (3.2%) (1–3 SRS treatments)
WBRT alone:	1 (1.6%)
Surgical resection + WBRT:	2 (3.2%)

ALK: anaplastic lymphoma kinase, EGFR: epidermal growth factor receptor, KRAS: Kirsten rat sarcoma viral oncogene homolog, PD: programmed death receptor, PDL: programmed death ligand, PBRT: partial brain radiotherapy, SRS: stereotactic radiosurgery and WBRT: whole brain radiotherapy.

## Data Availability

The datasets presented in this article are not readily available because use was only granted by the institutional review board for this specific study. Please contact the corresponding author for more details.

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
