# Peer review of "Contrast Clearance Analysis (CCA) to Assess Viable Tumour following Stereotactic Radiosurgery (SRS) to Brain Metastasis in Non-Small Cell Lung Cancer (NSCLC)"

_cancers, 2024, doi:10.3390/cancers16061218_

Round 1

Reviewer 1 Report (Previous Reviewer 1)

Comments and Suggestions for Authors

I am satisfied with the revised manuscript and point-to-point response. 

Author Response

Thank you very much

Reviewer 2 Report (Previous Reviewer 3)

Comments and Suggestions for Authors

The authors discuss an all too common problem that plagues those of us treating patients with brain metastasis, radiation necrosis (RN).  Thus far there are no single test that is effective in distinguishing RN from progression.  CCA is a simple approach that has shown high specificity and sensitivity in detection of RN.  The edits are much improved and concise.  The only minor question to address is what is the lower limit of detection for CCA.  We know that MR perfusion is ineffective with small volumes.  Lastly, authors should comment on how  CCA be affected in the setting of immunotherapy.   

Comments on the Quality of English Language

Minor edits on grammar as there are some run-on sentences

Author Response

we have added the following text to the discussion section: "The cost implication of CCA implementation will subject to further analyses but, as a provisional estimate, local costs of CCA are around £700 with FDG PET cost of over £2000 and MR perfusion around £1000".  

This manuscript is a resubmission of an earlier submission. The following is a list of the peer review reports and author responses from that submission.

Round 1

Reviewer 1 Report

Comments and Suggestions for Authors

I am very interested in learning utilizing CCA to assess viable tumors following stereotactic radiosurgery for NSCLC brain metastases. As the authors pointed out, it is crucial for a clinician treating brain metastases to have the confidence differentiating viable tumors from treatment effects following SRS. However, from the current study, it is difficult to appreciate it. As a gold standard (pathology diagnosis), out of 62 patients evaluated in the study, the authors presented only 4 patients who underwent surgical resection and had histology findings, although the data did not reveal any comparison between pathology findings and CCA and conventional MRI immediately prior to surgery. Table 1 should be included in the patients and methods section, rather than in the results portion. Besides, the presentation of Table 1 is not aligned. There is a redundant ALK rearrangement, I believe one should be EGFR. Due to the lack of pathology findings, there are no analyses of sensitivity and positive predictive value. Meanwhile, there is no description as to how to perform CCA. There is no study of patient survival, which better understand this cohort of patients. Most important, there are no statistical analyses to convince the readers regarding their conclusions. 

Author Response

Dear Reviewer,

Thank you for considering our revised manuscript. We are grateful to you for taking the time to provide this useful feedback. Please find our responses (green) to your comments (black) below.

I am very interested in learning utilizing CCA to assess viable tumors following stereotactic radiosurgery for NSCLC brain metastases. As the authors pointed out, it is crucial for a clinician treating brain metastases to have the confidence differentiating viable tumors from treatment effects following SRS. However, from the current study, it is difficult to appreciate it. As a gold standard (pathology diagnosis), out of 62 patients evaluated in the study, the authors presented only 4 patients who underwent surgical resection and had histology findings, although the data did not reveal any comparison between pathology findings and CCA and conventional MRI immediately prior to surgery.

Thank you for your comment. As you say, the comparison of CCA findings and histology is limited by the small numbers of patients undergoing resection and by other factors such as experience in analysis and timing of CCA pre-operatively and these have been clarified in the text of the manuscript.

Table 1 should be included in the patients and methods section, rather than in the results portion. Besides, the presentation of Table 1 is not aligned.

The table has been moved to the Methods section. Unfortunately, we are not able to edit the table alignment as this is decided by the Journal formatting template.

There is a redundant ALK rearrangement, I believe one should be EGFR.

The first line should have read “EGFR mutation” and has now been corrected.  

Due to the lack of pathology findings, there are no analyses of sensitivity and positive predictive value.

Given the small numbers of patients undergoing resection no sensitivity analysis was performed for this group. This limitation is discussed in the manuscript with addition of the sentence: “Also, the small number of patients undergoing surgical resection of brain lesions limits the correlation of CCA findings with histology.”  However, the clinical data suggest the correct CCA findings of radiation change based on subsequent scans and this has been added to the text.

Meanwhile, there is no description as to how to perform CCA.

A description of how CCA is performed, with references signposting to further detail, is included in the introduction (lines 75-82).

There is no study of patient survival, which better understand this cohort of patients.

Thank you. We have added the survival data to the manuscript in the Results “Contrast Clearance Analysis” section and added a short comment in the discussion (para 4). Interestingly the survival of those with disease progression on the first CCA scan appears shorter than the group as a whole, again supporting CCA showing the correct diagnosis in the absence of histological correlation.

Most important, there are no statistical analyses to convince the readers regarding their conclusions. 

Please see above responses.

Reviewer 2 Report

Comments and Suggestions for Authors

This is a novel technique to differentiate pseudo-tumor (radio-necrosis) from tumor progression and has a clinical significance. Although the design is retrospective, it adds to the literature and will eventually initiate RCT's in the future.

Author Response

Dear Reviewer,

Thank you for your kind comments on the manuscript.

Reviewer 3 Report

Comments and Suggestions for Authors

The authors seek to address a challenging and pertinent question for many of us in the neuro-oncology world-determining the differences between progressive disease and radiation necrosis.  While this is the largest data of NSCLC patients undergoing CCA, there are many voids in the data that need to be filled.

Introduction- discussion of treatment options for radionecrosis, vitamin E and pentoxifylline have shown some efficacy and should be mentioned.

Results-Under molecular profile, ALK rearrangement is listed twice.  Is this an error or does this represent two separate fusion proteins?  If so, please list.

Results-The timing between the radiation completion and the CCA scan should be documented as this can be suggestive of etiology.

Results-For the patient on page 4 lines 127-131, the second CCA scan suggested radionecrosis while the third scan suggested progressive tumor.  This raises the possibility that the CCA was insensitive in identifying early progressive tumor.

Results-For the patients who had CCA suggested progressive tumor growth, surgical confirmation showed 25% accuracy with 75% misdiagnosed.

Result-Page 5 line 143-145, the patient who had bevacizumab who had improvement in imaging appearances, does not preclude the possibility of progressive disease.  Bevacizumab can decrease overall enhancement and progressive tumor and radiation necrosis

Discussion-The authors suggest that the lack of concordance between CCA findings and those who underwent surgical resection were related to timelines longer than 14 days.  There are reported median timing between CCA and surgery was 30 days.  In their discussion, they proposed that pre-surgery CCA should be performed within 7 days.  How did they derive these numbers.

Are there any comparisons between CCA and MR perfusion or MR spectroscopy scans?

Long-term outcomes data, including whether the patient's succumbed to systemic disease or CNS disease would help validate the CCA findings.

Comments on the Quality of English Language

Minor editing of run-on sentences.

Author Response

Dear Reviewer,

Thank you for considering our revised manuscript. We are grateful to you for taking the time to provide this useful feedback. Please find our responses (green) to your comments (black) below.

The authors seek to address a challenging and pertinent question for many of us in the neuro-oncology world-determining the differences between progressive disease and radiation necrosis.  While this is the largest data of NSCLC patients undergoing CCA, there are many voids in the data that need to be filled.

Introduction- discussion of treatment options for radionecrosis, vitamin E and pentoxifylline have shown some efficacy and should be mentioned.

These have been added to the sentence describing treatment options.

Results-Under molecular profile, ALK rearrangement is listed twice.  Is this an error or does this represent two separate fusion proteins?  If so, please list.

The first line should have read “EGFR mutation” and has now been corrected. 

Results-The timing between the radiation completion and the CCA scan should be documented as this can be suggestive of etiology.

This data has now been added to the manuscript in the results – “Contract Clearance Analysis” section.  The timings were similar between the 2 groups.

Results-For the patient on page 4 lines 127-131, the second CCA scan suggested radionecrosis while the third scan suggested progressive tumor.  This raises the possibility that the CCA was insensitive in identifying early progressive tumor.

We have added language to express this important point in the discussion section. “Furthermore, in all patients whose initial CCA showed post-SRS changes and who underwent serial CCAs, the initial diagnosis was upheld with the serial imaging; two patients later developed progression which may be due to absence of progression at early time points though the possibility that CCA was less sensitive at assessing early disease progression cannot be excluded.”

Results-For the patients who had CCA suggested progressive tumor growth, surgical confirmation showed 25% accuracy with 75% misdiagnosed.

Thank you. The limitation of this have been expanded on in the discussion. In addition we have added new wording in the simple summary, abstract and results section and a new figure to expand on potential reasons why the poor concordance was seen. We feel that the addition of the discussion of the “thin blue rim” sign would be beneficial to those setting up their own CCA service. The patient included in the new figure has consented to inclusion of the imaging. A new author inputted to the new analyses so have been included in an amended authorship list.

Result-Page 5 line 143-145, the patient who had bevacizumab who had improvement in imaging appearances, does not preclude the possibility of progressive disease.  Bevacizumab can decrease overall enhancement and progressive tumor and radiation necrosis.

Thank you. The patient did not subsequently develop disease progression. We have added language to clarify.

Discussion-The authors suggest that the lack of concordance between CCA findings and those who underwent surgical resection were related to timelines longer than 14 days.  There are reported median timing between CCA and surgery was 30 days.  In their discussion, they proposed that pre-surgery CCA should be performed within 7 days.  How did they derive these numbers.

This is based on discussions with the team at Sheba hospital who developed CCAs – this has been added as a personal communication

Are there any comparisons between CCA and MR perfusion or MR spectroscopy scans?

We did not do this here but it is the subject of a prospective study about to start at our institution.  2 relevant studies have been added to the manuscript– one is from the Zach L et al who compared CCA with MRI perfusion, the other is a paper from Peker which also compares CCA with MRI perfusion.  However, in the Pecker paper a 2-year local control rate of metastases post SRS is reported as 98.9% and therefore their findings require further confirmation.

Long-term outcomes data, including whether the patient's succumbed to systemic disease or CNS disease would help validate the CCA findings.

Thank you. We have added the survival data to the manuscript in the Results “Contrast Clearance Analysis” section and added a short comment in the discussion (para 4). Interestingly the survival of those with disease progression on the first CCA scan appears shorter than the group as a whole, which is broadly supportive of the CCA results but, given the retrospective nature of the data has not been commented on further in the manuscript. No patients died of progressive brain disease.